# Mucosal effects of tenofovir 1% gel

Florian Hladik[1,2,3]*, Adam Burgener[4,5†], Lamar Ballweber[3†], Raphael Gottardo[3,6,7], Lucia Vojtech[1], Slim Fourati[8], James Y Dai[6,7], Mark J Cameron[8], Johanna Strobl[3], Sean M Hughes[1], Craig Hoesley[9], Philip Andrew[10], Sherri Johnson[10], Jeanna Piper[11], David R Friend[12], T Blake Ball[4,5], Ross D Cranston[13,14], Kenneth H Mayer[15], M Juliana McElrath[2,3,16], Ian McGowan[13,14]*

[1]Department of Obstetrics and Gynecology, University of Washington, Seattle, United States; [2]Department of Medicine, University of Washington, Seattle, United States; [3]Vaccine and Infectious Diseases Division, Fred Hutchinson Cancer Research Center, Seattle, United States; [4]Department of Medical Microbiology, University of Manitoba, Winnipeg, Canada; [5]National HIV and Retrovirology Laboratories, Public Health Agency of Canada, Winnipeg, Canada; [6]Department of Biostatistics, University of Washington, Seattle, United States; [7]Public Health Sciences Division, Fred Hutchinson Cancer Research Center, Seattle, United States; [8]Vaccine and Gene Therapy Institute of Florida, Port Saint Lucie, United States; [9]Department of Medicine, University of Alabama, Birmingham, United States; [10]FHI 360, Durham, United States; [11]Division of AIDS, National Institute of Allergy and Infectious Diseases, National Institutes of Health, Bethesda, United States; [12]CONRAD, Eastern Virginia Medical School, Arlington, United States; [13]University of Pittsburgh School of Medicine, Pittsburgh, United States; [14]Microbicide Trials Network, Magee-Women's Research Institute, Pittsburgh, United States; [15]Fenway Health, Beth Israel Deaconess Hospital, Harvard Medical School, Boston, United States; [16]Department of Global Health, University of Washington, Seattle, United States

**\*For correspondence:** fhladik@fhcrc.org (FH); mcgowanim@mwri.magee.edu (IMG)

†These authors contributed equally to this work

**Competing interests:** The authors declare that no competing interests exist.

**Abstract** Tenofovir gel is being evaluated for vaginal and rectal pre-exposure prophylaxis against HIV transmission. Because this is a new prevention strategy, we broadly assessed its effects on the mucosa. In MTN-007, a phase-1, randomized, double-blinded rectal microbicide trial, we used systems genomics/proteomics to determine the effect of tenofovir 1% gel, nonoxynol-9 2% gel, placebo gel or no treatment on rectal biopsies (15 subjects/arm). We also treated primary vaginal epithelial cells from four healthy women with tenofovir in vitro. After seven days of administration, tenofovir 1% gel had broad-ranging effects on the rectal mucosa, which were more pronounced than, but different from, those of the detergent nonoxynol-9. Tenofovir suppressed anti-inflammatory mediators, increased T cell densities, caused mitochondrial dysfunction, altered regulatory pathways of cell differentiation and survival, and stimulated epithelial cell proliferation. The breadth of mucosal changes induced by tenofovir indicates that its safety over longer-term topical use should be carefully monitored.
Clinical trial registration: NCT01232803.

## Introduction

The HIV prevention field has invested considerable resources in testing the phosphonated nucleoside reverse transcriptase inhibitor (NRTI) tenofovir as a mucosally applied topical microbicide to prevent sexual HIV transmission. In a phase 2B trial, CAPRISA 004, pericoital tenofovir 1% gel was 39% efficacious in preventing vaginal HIV acquisition (*Abdool Karim et al., 2010*). However, in another phase 2B trial, the VOICE study (MTN-003), the daily vaginal tenofovir 1% gel arm was discontinued for futility (*Marrazzo et al., 2015*). Adherence to product use was low in VOICE, likely explaining the

**eLife digest** Tenofovir is a drug that can stop some viruses—including HIV—from multiplying. It is commonly used in multidrug therapies to control HIV infection. Clinical trials are underway to find out whether using the drug in the form of a gel applied to the vagina or rectum could be an effective way to prevent HIV transmission during sex.

Some of the clinical trials carried out so far have produced promising results. However, since the use of gels containing anti-viral drugs is a new strategy for HIV prevention, there are limited data available about the safety of these products. Previous studies have shown that the concentration of tenofovir in the vagina is much higher in individuals using the gel than in those taking the tablet form of the drug. These high concentrations could lead to unexpected effects on the health of the cells exposed to the gel.

Here, Hladik, Burgener, Ballweber et al. used a systems biology approach to look at the broad effects of tenofovir gel on tissue from the rectum. Tissue samples taken from the rectums of 15 patients who used tenofovir gel for seven days were compared with tissue samples taken from individuals who used a control gel that did not contain the drug or who did not use any gel.

Genes that regulate inflammation were suppressed in the rectal tissue from patients who used tenofovir, as were genes that help these tissues regenerate and produce energy. The tissue from these patients also contained more immune cells, suggesting that their local immune systems were more active. Additionally, Hladik, Burgener, Ballweber et al. observed changes that could potentially lead to the increased growth of cells.

Similar differences were also observed in vaginal cells that had been treated with tenofovir in the laboratory. These findings suggest that tenofovir delivered directly to the vagina or rectum may have unintentional local side effects. However, it is important to acknowledge that tenofovir gel has been evaluated in multiple studies that have not observed overt clinical adverse effects. Therefore, the implication of these findings is currently unclear and warrants further study.

differences in findings between the two studies. Currently, the CAPRISA 004 study is being repeated in a phase 3 trial (FACTS 001 study).

A reduced glycerin formulation of the vaginal tenofovir 1% gel for use as a rectal microbicide appears safe when evaluated by epithelial sloughing, fecal calprotectin, inflammatory cytokine mRNA/protein levels, and cellular immune activation markers (*McGowan et al., 2013*). However, because topical application of an antiretroviral drug to the mucosa is a novel prevention strategy without clinical precedent, we conducted a comprehensive systems biology assessment of tenofovir gel's effects on the mucosa.

## Results

### Tenofovir 1% gel induces broad and pronounced gene expression changes in the rectum

We measured mRNA expression changes across the complete human transcriptome by microarrays in rectal biopsies taken at 9 and 15 cm proximal to the anal margin. Biopsies were obtained before treatment, after a single and after seven consecutive once-daily applications of reduced glycerin tenofovir 1% gel, nonoxynol-9 (N-9) 2% gel, hydroxyethyl cellulose (HEC) placebo gel, or no treatment (eight participants per arm were tested by microarrays). The primary results of the clinical study, MTN-007, a phase 1, randomized, double-blind, placebo-controlled trial at three US sites were reported elsewhere (*McGowan et al., 2013*). Relative to enrollment biopsies, after 7 days of treatment, tenofovir 1% gel suppressed 505 genes and induced 137 genes in the 9 cm biopsies, whereas the detergent N-9, a transient mucosal toxin, suppressed 56 genes and induced 60 genes (log$_2$ fold expression change $\geq$ 0.5 for induction or $\leq$ −0.5 for suppression, FDR $\leq$ 0.05) (*Figure 1A, B* and *Supplementary file 2*). 15 suppressed and 4 induced genes were common to tenofovir and N-9 (*Figure 1B*). In the HEC gel and no treatment arms, 16 and 23 genes changed (*Figure 1B*). Tenofovir 1% gel affected more genes after 7 days of treatment than after a single application and more genes in 9 cm than in 15 cm biopsies (*Figure 1B*), with significant correlations between expression changes at 9 cm and 15 cm (suppression: Spearman rho = 0.4775 [95% CI:

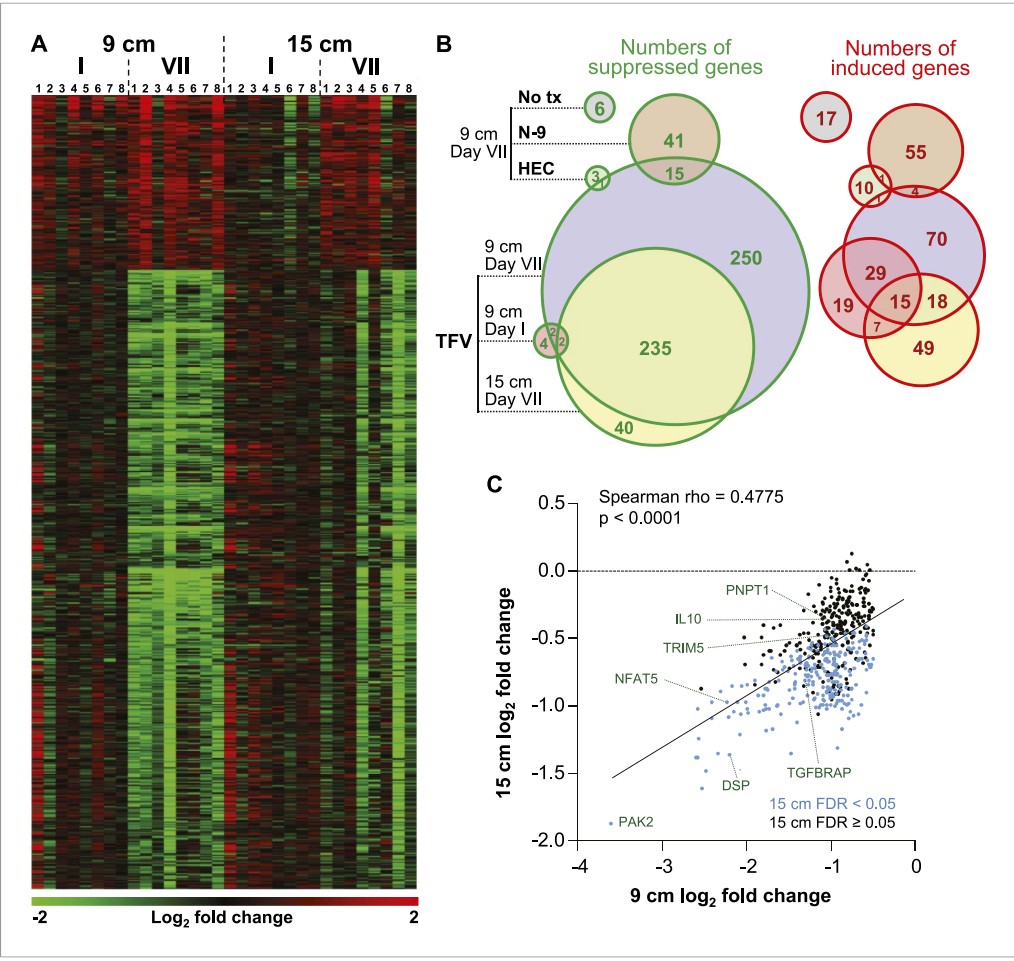

**Figure 1**. Tenofovir-induced gene expression changes in the human rectum. (**A**) Heat map of differentially expressed genes in eight participants after a single (I) and after seven consecutive once-daily (VII) rectal applications of tenofovir 1% gel compared to baseline in biopsies taken at 9 cm and 15 cm proximal to the anal margin. Red and green bars signify strength of gene induction and suppression, respectively. 642 genes are shown, all of which exhibited an estimated FDR $\leq$ 0.05 and a $\log_2$ fold expression change of $\geq$ 0.5 (induction) or $\leq$ −0.5 (suppression) when evaluated jointly for all eight participants at time point VII in the 9 cm biopsies. (**B**) Numbers of significantly suppressed (green borders and numbers) and induced (red borders and numbers) genes after reduced glycerin tenofovir 1% gel (TFV) treatment and their overlap with nonoxynol-9 (N-9), hydroxyethyl cellulose (HEC) and no treatment (No tx). Circle area symbolizes the number of affected genes, overlap the number of genes independently affected by two or three conditions. (**C**) Correlation of $\log_2$ fold gene suppression from baseline to Day VII between 9 and 15 cm biopsies. All 505 genes significantly suppressed at 9 cm are included. Genes depicted as blue dots were significantly suppressed in both 9 and 15 cm biopsies (15 cm FDR < 0.05), genes depicted as black dots were only significant in 9 cm biopsies (15 cm FDR $\geq$ 0.05). Spearman rho correlation between the 9 and 15 cm biopsies expression and the corresponding p-value of a Spearman rank correlation test are indicated on the plot. Genes tested in *Figure 2B* by RT-ddPCR are specifically indicated.

The following figure supplements are available for figure 1:

**Figure supplement 1**. Correlation of $\log_2$ fold gene induction by tenofovir 1% gel from baseline to Day VII between 9 and 15 cm biopsies.

**Figure supplement 2**. $\log_2$ fold gene suppression (**A**) and induction (**B**) from baseline to Day VII caused by N-9 and tenofovir in 9 cm biopsies.

0.4051–0.5440]; p < 0.001, *Figure 1C*; induction: Spearman rho = 0.427 [95% CI: 0.2840–0.5514]; p < 0.001, *Figure 1—figure supplement 1*). By fold change of the individual genes, tenofovir suppressed genes more strongly than N-9 (median 0.505 vs 0.627-fold; p < 0.001), but induced genes less strongly (1.58 vs 1.69-fold; p < 0.001) (*Figure 1—figure supplement 2*).

## Confirmatory RT-ddPCR quantification and in situ immunostaining in additional study participants

To independently confirm the microarray results, we selected nine induced and six suppressed genes and performed reverse transcription digital droplet PCR (RT-ddPCR) assays with RNA from the 9 cm biopsies in the remaining seven individuals enrolled in the tenofovir 1% gel arm (two men and five women), whose biopsies had not been analyzed by microarray (*Figure 2*). The mRNA copy numbers of all 15 genes increased or decreased as predicted from the microarray data between baseline and time point VII (*Figure 2A,B*). Next, we combined the microarray and RT-ddPCR expression data, normalized fluorescence and copy number values over their respective baselines, and compared the fold change after 7 days of treatment with baseline (*Figure 2C*). Expression changes for all 15 genes assessed by microarray and RT-ddPCR were statistically significant (p < 0.01 for all genes except TRIM5 [p = 0.02]).

For additional confirmation, we selected three induced (CD7, CD3 and ubiquitin D) and one suppressed gene (IL-10) for immunohistochemical staining of the respective proteins in 9 cm rectal biopsies from 10 subjects (*Figure 2D* and *Figure 2—figure supplements 1, 2*). Consistent with the gene expression studies, infiltrating T lymphocytes increased twofold to fivefold in the mucosa (mean fold change 5.0, 95% CI 2.46–10.1, p < 0.001 for CD7[+]; 2.44, 1.17–5.11, p = 0.023 for CD3[+]), whereas IL-10[+] columnar epithelial cells decreased by more than half (0.36, 0.18–0.79, p = 0.017), between baseline and following 7 days of tenofovir 1% gel use. Ubiquitin D was widely expressed in all biopsies, but tenofovir treatment increased the intensity of its expression (p = 0.007), as predicted by the microarrays.

## Gene expression patterns and functional pathways

Tenofovir 1% gel was more suppressive than stimulatory, with a ratio of induced to suppressed genes in the 9-cm rectal biopsies of 0.116 (17 genes up-regulated to 146 down) for nuclear products. However, genes encoding secreted proteins were more often induced than suppressed (*Figure 3A*), with a ratio of 2.33 (35 genes up-regulated to 15 down; $\chi^2$ p < 0.001). Noteworthy among induced genes for secreted products were the chemokines CCL2, CCL19, CCL21, CCL23, CXCL9, and CXCL13 (*Figure 3A*). Correspondingly, transcripts of a number of leukocyte-specific cell surface markers increased, specifically CD2, CD3D, CD7, CD8A, CD19, CD52, CD53, CCR6, and CCR7. The kinetics of gene induction is depicted in *Figure 3—figure supplement 1*.

Among suppressed genes with products localizing to the cell nucleus, we identified a large number of known or putative transcription factors and their co-factors, including CREB1 (CAMP responsive element-binding protein 1) and CREBBP (CREB-binding protein), both activators of IL-10 transcription (*Woodgett and Ohashi, 2005*; *Alvarez et al., 2009*), NFAT5 (nuclear factor of activated T cells 5) (*Neuhofer, 2010*), and many zinc finger proteins (*Figure 3A*). Tenofovir 1% gel also suppressed genes important for regulation of transcription and translation, as well as biological processes involving transforming growth factor beta (TGF-β), epithelial structure organization, regulation of cell proliferation, and apoptosis (*Figure 3—figure supplement 2*). Lastly, tenofovir 1% gel suppressed genes important for mitochondrial function, including PNPT1 (polyribonucleotide nucleotidyltransferase 1) (*Wang et al., 2010*) and OPA3 (optic atrophy 3) (*Misaka et al., 2002*). Among the few genes with nuclear products that were strongly induced, KIAA0101 and ubiquitin D were notable (*Yu et al., 2001*; *Lee et al., 2003*; *Hosokawa et al., 2007*).

## Similarities between vaginal keratinocytes and rectal biopsies

Tenofovir 1% gel is most advanced as a candidate product for vaginal HIV prevention. We were therefore interested to extend our findings from the rectum to the vaginal mucosa. We isolated epithelial cells from vaginal tissues donated by three healthy women and tested their response to 50 and 500 μM tenofovir in vitro for up to 14 days of culture using RNA expression analysis. Preliminary tests showed that these dosages give roughly similar intracellular concentrations of the active drug (tenofovir diphosphate) as measured in the vagina after tenofovir 1% gel use (not shown)

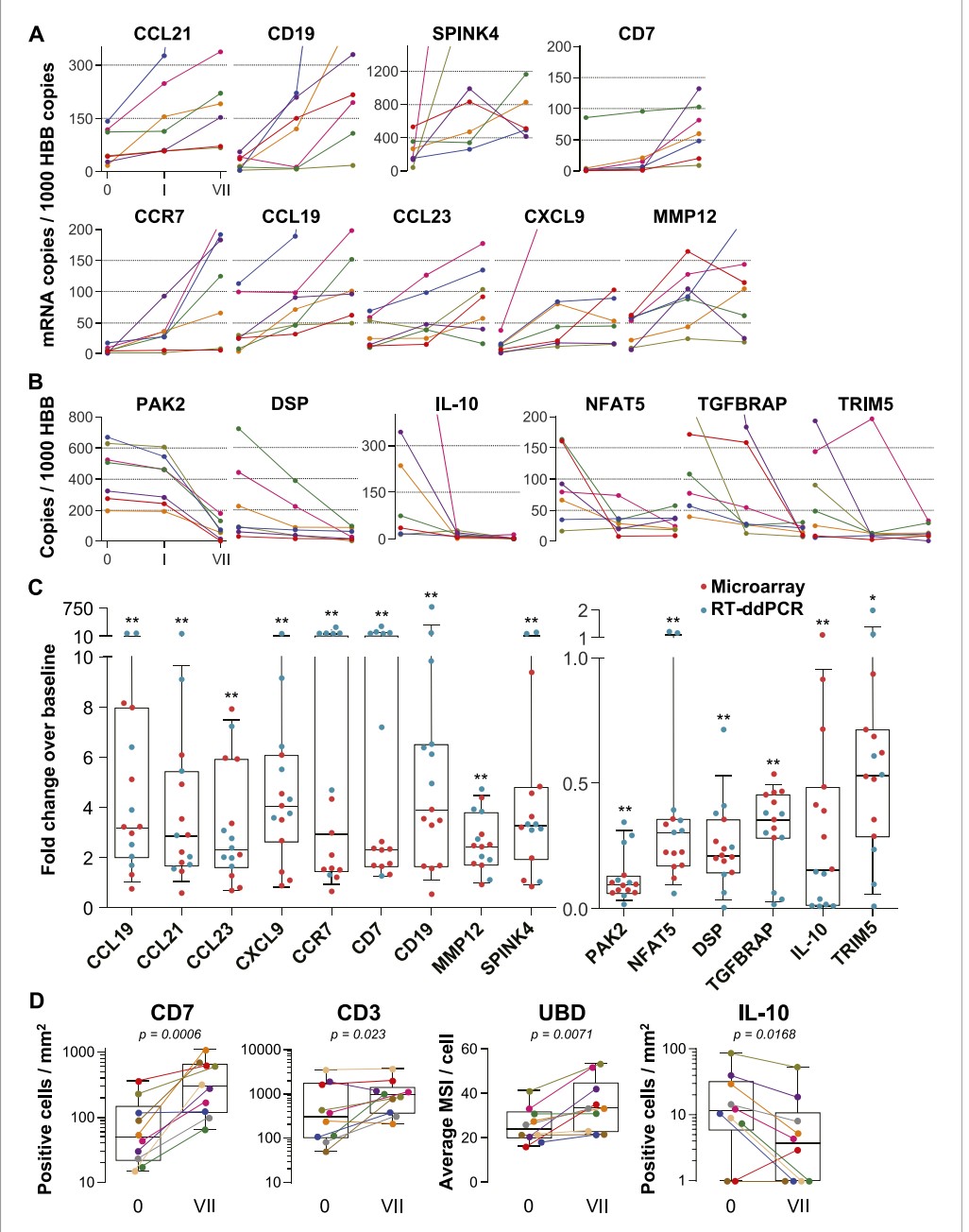

**Figure 2**. Confirmation of microarray data. (**A** and **B**) Quantification of mRNA copy numbers measured in 9 cm biopsies by reverse transcription (RT) droplet digital PCR (ddPCR) relative to the housekeeping gene hemoglobin beta (HBB) copy numbers in seven additional study participants. (**A**) Nine selected genes induced in the microarrays: CCL19, CCL21, CCL23, CXCL9, CCR7, CD7, CD19, matrix metallopeptidase 12 (MMP12), and serine peptidase inhibitor of the Kazal type 4 (SPINK4). Copy numbers at baseline (0), after a single tenofovir gel application (I) and after seven consecutive once-daily applications (VII) are shown. Line colors signify each of the seven study participants. (**B**) Six selected suppressed genes: p21-activated kinase (PAK2), nuclear factor of activated T cells 5 (NFAT5), desmoplakin (DSP), TGF-β receptor associated protein (TGFBRAP), interleukin 10 (IL-10), and tripartite motif-containing protein 5 (TRIM5). (**C**) Normalized fold changes of gene expression at Day VII over baseline in all 15 individuals treated with tenofovir 1% gel. Red dots depict fold changes measured by microarray, blue dots depict fold changes measured by RT-ddPCR. The boxes indicate median and 25th–75th percentiles and the whiskers indicate 10th–90th percentile. Asterisks indicate statistical significance level relative to baseline (one asterisk p < 0.05; two asterisks p < 0.01, by one-sided Wilcoxon tests, adjusted for multiple testing). (**D**) Immunostaining of formalin-fixed 9 cm rectal biopsies from 10 participants for the proteins CD7 (immunohistochemistry [IHC]), CD3

*Figure 2. continued on next page*

*Figure 2. Continued*

(immunofluorescence) and ubiquitin D (UBD; IHC), predicted to be induced by the microarrays, and for IL-10 (IHC), predicted to be suppressed. For CD7 and CD3, tissue sections were evaluated in their entirety and positive cells per $mm^2$ are shown at baseline (0) and after seven consecutive once-daily applications (VII). Representative images are shown in *Figure 2—figure supplement 1*. For UBD and IL-10, only columnar epithelial cells were evaluated. For UBD, the average mean staining intensities (MSI) per cell are shown. Representative images are shown in *Figure 2—figure supplement 2*. Colors signify each of the 10 study participants. The boxes indicate median and 25th–75th percentiles and the whiskers indicate the range. Paired Wilcoxon signed-rank test p values for differences between 0 and VII are listed.

The following figure supplements are available for figure 2:

**Figure supplement 1**. Immunohistochemistry for CD7, and immunofluorescence for CD3, in rectal biopsies before (0) and after 7 days (VII) of daily tenofovir 1% gel use.

**Figure supplement 2**. Immunohistochemistry for IL-10 and ubiquitin D (UBD) in rectal biopsies before (0) and after 7 days (VII) of daily tenofovir 1% gel use.

---

(*Hendrix et al., 2013*). Pre-processed microarray expression data were extremely consistent between the three vaginal cell cultures (mean Pearson correlation coefficient 0.9912; *Figure 3—source data 1*). Tenofovir's effects on the purified vaginal epithelial cells (*Figure 3B*) were similar to its effects on the rectal mucosa (*Figure 3A*), but, as expected, changes likely driven by leukocytes in the biopsies were not seen in the purified epithelial cells, or were differently regulated. In the epithelial cells, tenofovir did not significantly change chemokine, chemokine receptor, and cluster of differentiation (CD) genes, and it suppressed ISG 15 (interferon-stimulated gene 15) and MX1 (myxovirus resistance 1), which were induced in the biopsies.

The number of genes affected was initially higher with 500 µM than with 50 µM tenofovir, but equalized after 14 days of culture (*Figure 4A*). Next, we confirmed the expression changes of select genes by ddPCR with vaginal epithelial cells from four healthy women: mRNA copies of DSP (desmoplakin) and IL-10 significantly decreased, and KIAA0101 significantly increased during 7 days of tenofovir treatment, as seen in the microarray data (*Figure 4B*). In fact, IL-10 transcripts were virtually eliminated at 7 days (p = 0.002 and p = 0.003 for 50 and 500 µM, respectively), and KIAA0101 increased more than 10-fold at 500 µM tenofovir (p = 0.005). By ELISA of cell lysates, IL-10 protein also decreased significantly (n = 4 cell lines; p = 0.007 and p < 0.001 for 50 and 500 µM, respectively) (*Figure 4C*).

Tenofovir treatment of primary vaginal epithelial cells in vitro mostly impacted the same biological processes as it did in the rectum in vivo (*Figure 4D* and *Figure 3—figure supplement 2*). Additionally, it suppressed genes important for keratinocyte differentiation and cellular innate immunity, and induced genes involved in DNA damage repair. Furthermore, tenofovir enhanced vaginal epithelial cell proliferation and/or cell survival in vitro (p = 0.02) (*Figure 4E*).

## Signs of mitochondrial dysfunction

Our microarray results indicated that tenofovir suppresses PNPT1 (*Figure 1C*), which has been characterized as a master regulator of RNA import into mitochondria and whose deletion impairs mitochondrial function (*Wang et al., 2010*). To explore tenofovir's effects on mitochondria, we first confirmed its inhibition of PNPT1 in the 9 cm and 15 cm rectal biopsies of all 15 study participants by RT-ddPCR. PNPT1 copy numbers decreased more than 10-fold at 9 cm (mean fold change 0.09, 95% CI 0.009–0.172, p < 0.001) and by half at 15 cm (0.53, 0.34–0.73, p < 0.001) after 7 days of treatment (*Figure 5A*). To directly assess mitochondrial function, we picked one of the 13 genes encoded by mitochondrial DNA, ATP synthase F0 subunit 6 (ATP6), a key component of the proton channel (*Houstek et al., 2006*), and measured its transcription by RT-ddPCR in the 9 cm biopsies of all 15 study participants in the tenofovir arm (*Figure 5B*). Because mitochondrial genes are not included on the microarray chips, we had no preexisting information on its expression. ATP6 mRNA copy numbers decreased on average threefold (p = 0.003) after a single application and sixfold after 7 days (p < 0.001). In contrast, ATP6 copy numbers were stable in all 15 study participants treated with 2% N-9 gel (p = 0.491) (*Figure 5B*).

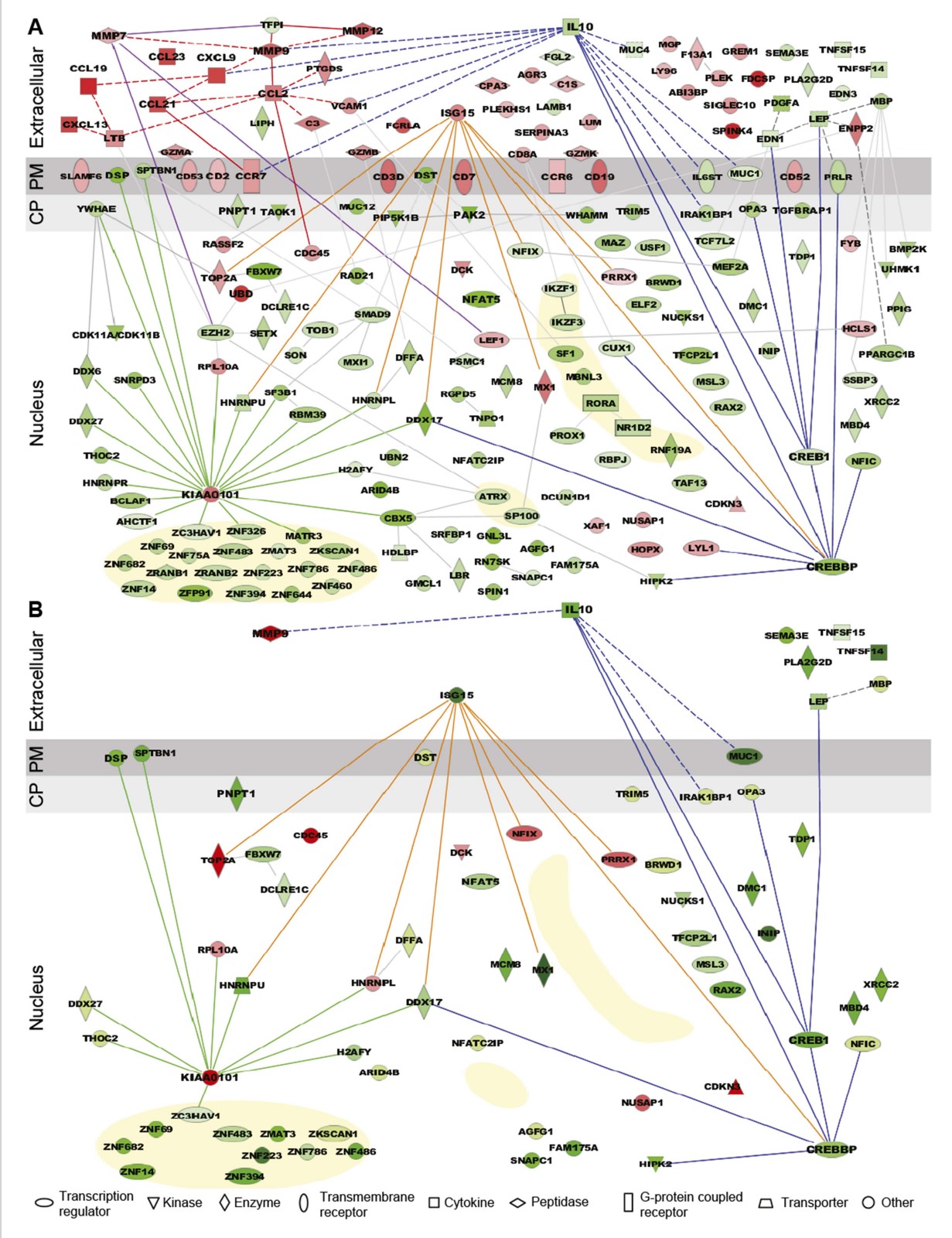

**Figure 3**. Expression pattern and functional pathway analysis. (**A**) Ingenuity pathways analysis of tenofovir-induced effects in rectal biopsies, showing cellular localizations of and relationships between individual gene products. Red symbols indicate induction and green symbols suppression at Day VII relative to baseline in 9 cm biopsies. The diagram includes all significant genes identified as primarily located in the extracellular space and the cell

*Figure 3. continued on next page*

*Figure 3. Continued*

nucleus. A few selected significant genes with products localizing to the plasma membrane (PM) or cytoplasm (CP) are also shown based on their putative functional roles. Direct (solid lines) and indirect (dashed) interactions between gene products are indicated. Line color is arbitrary and meant to indicate relationships between groups of genes. Yellow-shaded areas indicate zinc finger transcription factors. (**B**) Pathways analysis of tenofovir-induced effects in primary vaginal epithelial cells. Only genes that were suppressed or induced by tenofovir both in 9 cm rectum in vivo and in vaginal epithelial cells in vitro are shown. Primary vaginal epithelial cells derived from three healthy women were cultured with 50 or 500 μM tenofovir for 14 days. Global gene expression microarrays at 4, 7, and 14 days of culture were evaluated in comparison to untreated epithelial cells. Pre-processed microarray expression data were extremely consistent between the three vaginal cell cultures (mean Pearson correlation coefficient 0.9912; *Figure 3—source data 1*).

The following source data and figure supplements are available for figure 3:

**Source data 1**. Pearson correlation coefficients of pre-processed microarray probe expression values between the three primary vaginal cell cultures.

**Figure supplement 1**. Average strength of gene induction by tenofovir 1% gel across all 8 microarray study participants.

**Figure supplement 2**. Selected biological processes defined in the InnateDB database with significant enrichment of genes suppressed or induced by tenofovir 1% gel at Day VII in 9 cm biopsies.

Next, we evaluated changes in mitochondrial number and size between baseline and after 7 days of treatment in two study participants chosen for exhibiting pronounced PNPT1 suppression by tenofovir. The number of mitochondria per $\mu m^2$ decreased by more than half after 7 days of tenofovir treatment ($p < 0.001$ for both subjects) (*Figure 5C*). In the second participant, mitochondria also increased in size by 1.4-fold ($p < 0.001$) (*Figure 5D*) and developed dysmorphic cristae during treatment (*Figure 5E*). In parallel, tenofovir also caused statistically significant inhibition of PNPT1 and ATP6 mRNA expression in vaginal epithelial cells (*Figure 5F*).

## Proteomics of rectal secretions corroborates enhancing effect of tenofovir on cell survival

Extensive protein studies were not an intended part of MTN-007 and samples were not preserved optimally for this purpose. However, 483 proteins were detected consistently at baseline and time point VII by mass spectrometry. 382 proteins increased in the tenofovir arm vs 112 in the no treatment arm (five participants/arm). Among these, significant increases in individual protein expression were only seen in the tenofovir arm (*Figure 6*). The top 100 proteins exhibited an average fold increase of 3.8 (median 3.2, range 1.6–128.7) in the tenofovir arm, listed by p value in *Figure 6*. Enrichment analysis primarily indicated enhancement of cell survival and induction of leukocyte migration by tenofovir (*Figure 6*), which is consistent with our other findings.

## Discussion

Our findings indicate that reduced glycerin rectal tenofovir 1% gel affects expression of a different and much broader range of genes than N-9 2% gel, potentially affecting mucosal immune homeostasis, mitochondrial function, and regulation of epithelial cell differentiation and survival. These results make biological sense given that tenofovir is a DNA chain terminator, with possible off-target effects in human cells (*Lewis et al., 2003*), and that topical application achieves at least 100-fold higher active drug concentrations in the mucosa than oral administration of 300 mg tenofovir disoproxil fumarate (*Anton et al., 2012*; *Hendrix et al., 2013*). Moreover, tenofovir caused similar changes in primary vaginal epithelial cells cultured from several healthy women.

We did not find evidence that tenofovir directly causes inflammation, which is in keeping with our prior report that rectal tenofovir gel did not cause overt histological inflammation or increased mRNA/protein levels of a select panel of pro-inflammatory cytokines (*McGowan et al., 2013*). Rather, tenofovir dampened anti-inflammatory factors. Most prominently, it strongly inhibited IL-10 gene and protein expression, likely via blocking of CREB1 and its coactivator CREBBP (*Martin et al., 2005*; *Woodgett and Ohashi, 2005*; *Gee et al., 2006*; *Alvarez et al., 2009*). In addition, it suppressed signaling pathways downstream of TGF-β, a central anti-inflammatory mediator in the gut (*Konkel and Chen, 2011*; *Surh and Sprent, 2012*). Consequently, a number of chemokines were induced, such as

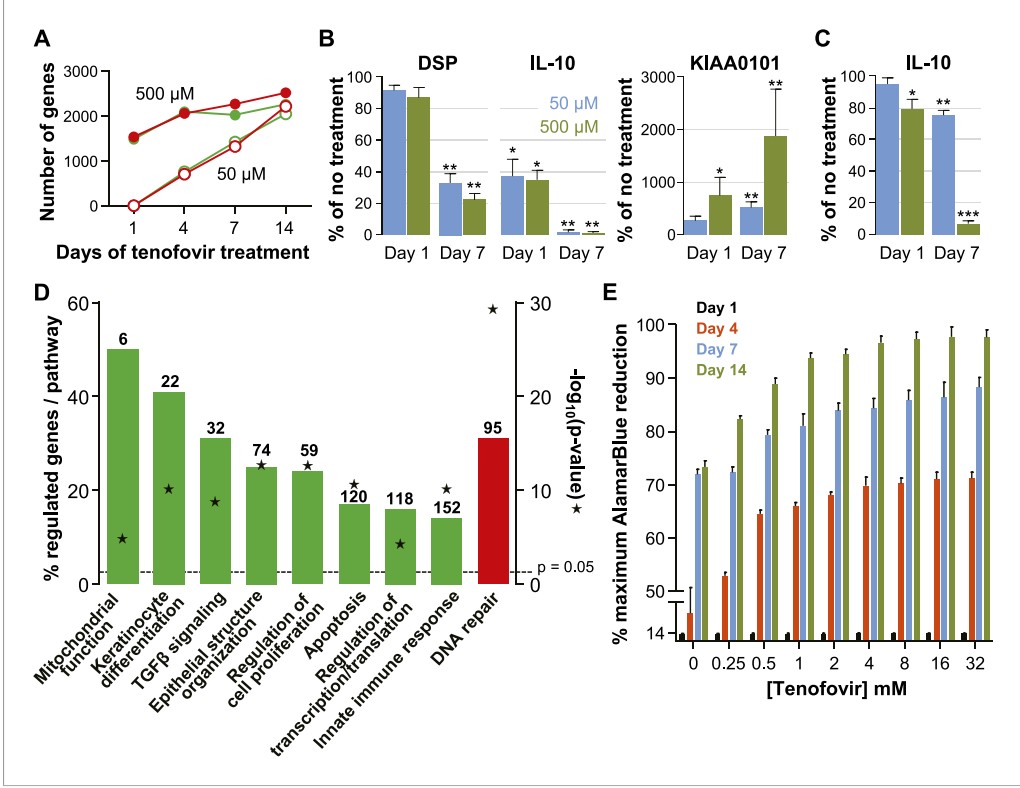

**Figure 4.** Effects of tenofovir on primary vaginal epithelial cells. (**A**) Number of suppressed (green) and induced (red) genes in response to treatment with 50 μM (open circles) or 500 μM (filled circles) tenofovir for 1, 4, 7, or 14 days (n = 3 cell lines from different women). (**B**) Quantification of mRNA copy numbers at days 1 and 7 of culture by RT-ddPCR assays for two selected genes identified as suppressed (DSP and IL-10) and one induced (KIAA0101) in the microarray data set (n = 4 cell lines). (**C**) Quantification of IL-10 protein concentrations in vaginal epithelial cells at days 1 and 7 of culture by ELISA. Mean (±standard deviation) IL-10 concentrations in the untreated cultures were 5.65 pg/ml (±0.25) at day 1 and 5.86 pg/ml (±0.32) at day 7 of culture. Boxes and error bars in (**B**) and (**C**) signify means and standard deviations with vaginal epithelial cell cultures derived from four healthy women. Asterisks indicate statistical significance level relative to untreated (*p < 0.05; **p < 0.01; ***p < 0.001). (**D**) Selected biological processes defined in the InnateDB and DAVID databases with significant enrichment of genes suppressed or induced by 50 μM tenofovir in vaginal epithelial cells after 7 days of culture. Green bars depict the percentage of genes identified as suppressed in a particular process out of the total number of genes included in that process. Red bars depict gene induction. Numbers of suppressed and induced genes are indicated above the bars. Gene enrichment in each biological process was tested for statistical significance as described in the 'Materials and methods' and the computed p values are depicted by the stars. Not all processes with significant gene enrichment are shown. (**E**) Proliferation of vaginal epithelial cells without or with various concentrations of tenofovir (n = 3 cell lines). Boxes depict mean percent reduction of the alamarBlue reagent in comparison to the maximum reduction. Error bars signify standard deviations.

the B lymphocyte chemoattractant CXCL13 (*Ansel et al., 2002*), and CCL19 and CCL21, both ligands of CCR7 on T lymphocytes and dendritic cells (*Forster et al., 2008*). Correspondingly, CCR7, the B cell marker CD19, and the T cell markers CD2, CD3D and CD7 increased. In keeping with this, we observed higher densities of CD3[+] and CD7[+] T lymphocytes in the rectal mucosa following 7 days of tenofovir 1% gel use. In concert, these changes suggest that tenofovir creates a state of potential hyper-responsiveness to external inflammatory stimuli but does not itself cause inflammation. In populations who, unlike our MTN-007 study cohort, have a high incidence of mucosal infections and associated immune activation, this could potentially diminish the anti-viral protective effect of topical tenofovir prophylaxis (*Naranbhai et al., 2012*).

Mitochondrial toxicity of nucleotide/nucleoside reverse transcriptase inhibitors such as tenofovir is well described but the mechanism remains unclear (*Lewis et al., 2003*). We found

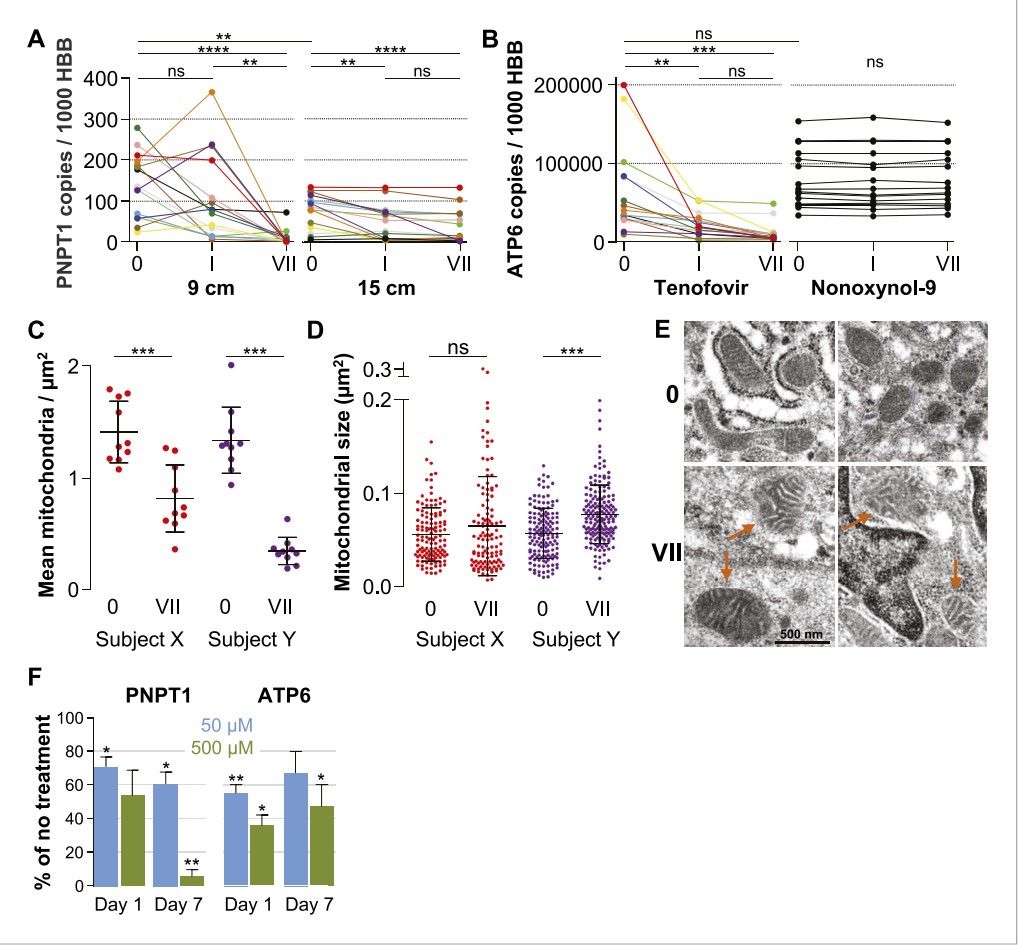

**Figure 5**. Quantification of mitochondria-associated parameters. (**A**) PNPT1 mRNA copy numbers measured in 9 and 15 cm biopsies at baseline (0), after a single tenofovir gel application (I) and after seven consecutive once-daily applications (VII) by RT-ddPCR assay. (**B**) Mitochondrial ATP6 mRNA copy numbers measured in 9 cm biopsies after tenofovir or N-9 treatment. Line colors in (**A**) and (**B**) signify the 15 participants in the tenofovir arm. Black lines signify the 15 participants in the N-9 arm. Baseline values were compared between 9 and 15 cm biopsies by paired t-test and between tenofovir and N-9 by unpaired t-test. Expression changes over time were tested for statistical significance by ANOVA with Bonferroni adjusted post-tests. (**C**) Assessment of mitochondrial density by electron microscopy of 9 cm biopsies in two study participants. Each dot indicates the mean number of mitochondria per $\mu m^2$ in a separate 2000× image. (**D**) Assessment of mitochondrial sizes by electron microscopy in the same biopsies. Each dot depicts the size in $\mu m^2$ of an individual mitochondrion measured at 5000×. Dot colors in (**C**) and (**D**) correspond to the line colors of the same two study participants in (**A**) and (**B**). Density and size changes were tested for statistical significance by unpaired t-tests. Horizontal lines and error bars depict means and standard deviations. (**E**) Representative electron microscopy images of normal mitochondria at baseline, and of enlarged and dysmorphic mitochondria at time point VII, in 9 cm biopsies of Subject Y. Fine structural detail is limited due to formalin fixation of biopsies. (**F**) PNPT1 and ATP6 gene expression in vaginal epithelial cell cultures in response to 1 and 7 days of 50 $\mu M$ (blue boxes) or 500 $\mu M$ (green boxes) tenofovir exposure in vitro. Boxes and error bars signify means and standard deviations across four independent experiments with epithelial cell cultures derived from the vaginal mucosa of four healthy women. Statistical significance levels in all figure panels are indicated by asterisks (*$p < 0.05$; **$p < 0.01$; ***$p < 0.001$; ****$p < 0.0001$; 'ns', not significant).

that tenofovir consistently inhibited expression of PNPT1, which encodes polynucleotide phosphorylase (PNPASE). PNPASE regulates nucleus-encoded RNA import into mitochondria (*Wang et al., 2010*). In PNPT1 knock-out mice, mitochondrial morphology and respiratory capacity are disrupted in a manner quite similar to the disruption in renal proximal tubular cells in patients with tenofovir-induced nephrotoxicity (*Perazella, 2010*; *Wang et al., 2010*). In our study,

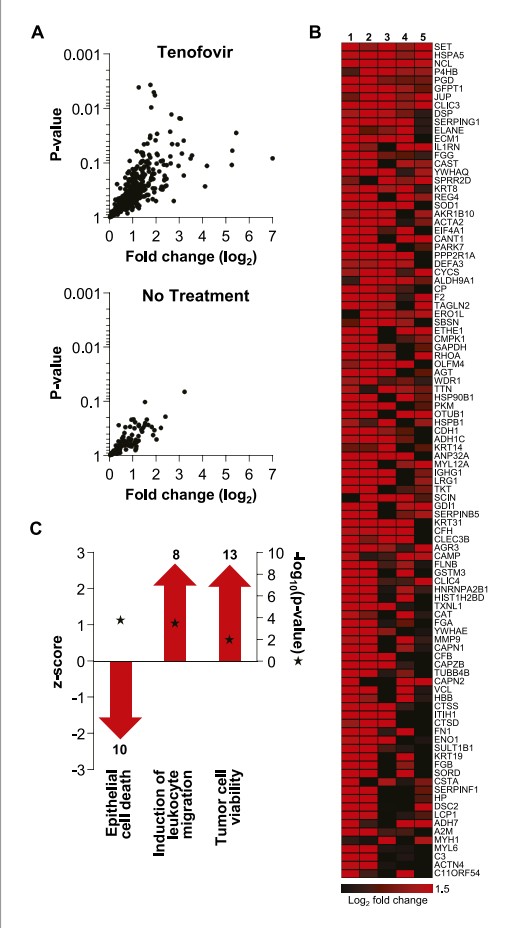

**Figure 6**. Unbiased mass spectrometry proteomics in rectal secretions from five study participants each in the tenofovir and the no treatment arm. (**A**) 483 proteins were consistently detected in all 10 study participants. Of these, 382 proteins in the tenofovir arm had $\log_2$ values > 0 for fold change between baseline and after seven daily gel applications, and 112 had $\log_2$ fold change values > 0 in the no treatment arm. $\log_2$ fold changes (x axis) and p values signifying the likelihood of change (y axis) for these proteins are shown. Upper panel, tenofovir arm (382 proteins); lower panel, no treatment arm (112 proteins). Data for proteins with $\log_2$ fold values ≤ 0 were not interpretable and are not shown. (**B**) $\log_2$ fold changes of the top 100 proteins upregulated between baseline and after 7 days of daily gel application in each of the five study participants in the tenofovir arm. (**C**) Selected biofunctional processes defined in the Ingenuity database with significant enrichment of proteins induced in rectal secretions by 7 days of daily tenofovir 1% gel use. The red bars with arrow heads depict the z scores for these biofunctions, indicating the strength of the directionality of the effect. Protein enrichment in each biofunction was tested for statistical significance as described in the 'Materials and methods' and the computed p values are depicted by the stars. Numbers of induced proteins are indicated above/below the bars.

just 1 week of daily tenofovir 1% gel application lowered transcription of mitochondrial ATP6 by sixfold and caused visible ultrastructural mitochondrial changes. These findings suggest that tenofovir's suppression of PNPT1 expression may underlie its reported, but heretofore unexplained, mitochondrial toxicity.

A number of changes in rectal biopsies and primary vaginal epithelial cells also suggested that tenofovir can cause increased epithelial proliferation. Furthermore, tenofovir's negative effect on mitochondrial function could lead to impairment of tumor progenitor cell apoptosis (*Modica-Napolitano et al., 2007*; *Ni Chonghaile et al., 2011*), as has been specifically reported for loss-of-function mutations of mtATP6, a mitochondrial gene strongly suppressed by tenofovir in our study (*Shidara et al., 2005*). Neoplastic pressure could also arise from the strong induction of KIAA0101 and UBD (ubiquitin D). KIAA0101 is important for regulation of DNA repair (*Simpson et al., 2006*), is increased in tumor tissues (*Yu et al., 2001*), and enhances cancer cell growth (*Jain et al., 2011*; *Hosokawa et al., 2007*). UBD appears to increase mitotic non-disjunction and chromosome instability (*Ren et al., 2006, 2011*) and is highly up-regulated in gastrointestinal cancers (*Lee et al., 2003*; *Ren et al., 2006, 2011*). Notably, though, these findings remain circumstantial, as there is no actual clinical evidence for carcinogenicity. Nevertheless, they raise the question of whether the relatively high concentrations of tenofovir achieved in the mucosa during topical use could potentially lead to neoplastic lesions with continuous and long-term use. According to Viread's Product Monograph, gastrointestinal tumorigenicity has been observed in mice after high oral dosing of tenofovir disoproxil fumarate. Vaginal tumorigenicity has been documented for azidothymidine, an NRTI and DNA chain terminator like tenofovir, which induced vaginal hyperplasia and carcinomas when delivered to mice intravaginally as a 2% solution (~25% carcinoma rate) (*Ayers et al., 1996*).

This is the first time that a systems biology approach has been applied to a clinical trial of mucosal pre-exposure prophylaxis, and our study shows the value of using these technologies for comprehensive mucosal safety assessment. Our findings raise concerns regarding the safety of topical tenofovir 1% gel in the rectum with long-term use. Tenofovir's effects on vaginal epithelial cells suggest similar activities in the vagina, which we are currently verifying in MTN-014, a phase I

clinical trial comparing vaginal and rectal tenofovir 1% gel in a cross-over format. Further studies are required to gauge whether tenofovir, which has become a valuable cornerstone drug in treating HIV infection, can also be safely and effectively used as a vaginal or rectal microbicide.

## Materials and methods

### Design of the clinical study

MTN-007 (ClinicalTrials.gov registration NCT01232803) was a phase 1, double blind, placebo-controlled trial in which participants were randomized to receive rectal reduced glycerin tenofovir 1%, nonoxynol-9 (N-9) 2% or hydroxyethylcellulose (HEC) gels, or no-treatment (1:1:1:1), at three clinical research sites (Pittsburgh, PA; Birmingham, AL; Boston, MA). The study protocol was approved by IRBs at all three sites. All participants gave written informed consent. The study details, and general safety and acceptability data, have been published elsewhere and showed that rectal tenofovir 1% gel was well tolerated and appeared safe by established safety parameters (*McGowan et al., 2013*). Each gel was administered as a single dose and then, after at least a 1-week recovery period, once daily for seven consecutive days. The first dose of study product was self-administered under supervision by the clinic staff at the Treatment 1 Visit. Subsequent administrations occurred at home, and study participants were instructed to insert one dose of gel into the rectum once daily throughout the 7-day period in the evening or before the longest period of rest.

### Study participants and products

A total of 65 study participants were enrolled and randomized in the study, 62 of whom completed it (tenofovir, *n* = 15; N-9, *n* = 16; HEC, *n* = 15; and no treatment, *n* = 16). 43 (69%) were male. Microarray studies were performed on eight randomly selected male participants in each group, and confirmatory gene expression studies were done on the remaining participants. The study population consisted of healthy, HIV-uninfected adults aged 18 or older who were required to abstain from receptive anal intercourse during the course of the clinical trial. Female participants were required to use effective contraception. Individuals with abnormalities of the colorectal mucosa, significant gastrointestinal symptoms (such as a history of rectal bleeding or inflammatory bowel disease), evidence of anorectal *Chlamydia trachomatis* or *Neisseria gonorrhea* infection, hepatitis B infection, or who used anticoagulants were excluded from the study. Reduced glycerin tenofovir 1% gel and HEC gel, known as the 'Universal Placebo Gel' (*Tien et al., 2005*), were supplied by CONRAD (Arlington, VA, USA). 2% N-9 gel was provided as Gynol II (Johnson & Johnson). All study products were provided in identical opaque HTI polypropylene pre-filled applicators (HTI Plastics, Lincoln, NE) containing 4 ml of study product.

### Mucosal biopsy procedures

Rectal biopsies for the microarray studies were obtained before treatment at enrollment (time point '0'), 30–60 min following application of the single gel dose (time point 'I'; to test acute single-dose effects), and again on the day following the last dose of the seven once-daily gel applications (time point 'VII'; to test multiple-dose effects). Following an enema with Normosol-R pH 7.4, a flexible sigmoidoscope was inserted into the rectum and biopsies were collected at 15 cm from the anal margin. Following the sigmoidoscopy, a disposable anoscope was inserted into the anal canal for collection of rectal biopsies at 9 cm from the anal margin. Immediately after harvest, biopsies were immersed in RNA later (Qiagen, Germany), stored at 4°C overnight, and transferred to a −80°C freezer for long-term storage until shipping to Seattle and processing.

### Primary vaginal keratinocyte cultures

Tissues routinely discarded from vaginal repair surgeries were harvested from four otherwise healthy adult women, placed in ice-cooled calcium- and magnesium-free phosphate-buffered saline containing 100 U/ml penicillin, 100 μg/ml streptomycin, and 2.5 μg/ml Fungizone (Thermo Fisher Scientific, Waltham, MA), and transported to the laboratory within 1 hr of removal from the donor. Tissue harvesting and experimental procedures were approved by the Institutional Review Boards of the University of Washington and the Fred Hutchinson Cancer Research Center. The deep submucosa was removed with surgical scissors and the remaining vaginal mucosa was cut into 5 × 5 mm pieces, which were incubated at 4°C for 18 hr in 5 ml of a 25 U/ml dispase solution (354235; BD Biosciences, Franklin Lakes, NJ). The epithelial sheets were dissected off under a stereoscope and incubated for 10–12 min at 37°C in 2 ml 0.05% trypsin while gently

shaking. The dispersed cells were poured through a 100-μm cell strainer into a 50-ml tube, pelleted by centrifugation, and resuspended in F medium (3:1 [vol/vol] F12 [Ham]-DMEM [Thermo Fisher Scientific], 5% fetal calf serum [Gemini Bio-Products, Calabasas, CA], 0.4 μg/ml hydrocortisone [H-4001; Sigma-Aldrich, St. Louis, MO], 5 μg/ml insulin [700-112P; Gemini Bio-Products], 8.4 ng/ml cholera toxin [227036; EMD Millipore, Billerica, MA], 10 ng/ml epidermal growth factor [PHG0311; Thermo Fisher Scientific], 24 μg/ml adenine [A-2786; Sigma], 100 U/ml penicillin, and 100 μg/ml streptomycin [Thermo Fisher Scientific]). The keratinocytes were plated into culture flasks in the presence of ~12,500/cm² irradiated (6000 Rad) 3T3-J2 feeder fibroblasts (a kind gift by Cary A Moody) and 10 μM of Rho kinase inhibitor Y27632 (1254; Enzo Life Sciences, Farmingdale, NY) was added (*Rheinwald and Green, 1975*; *Chapman et al., 2010*; *Liu et al., 2012*). Keratinocytes were fed every 2–3 days and passaged when around 80% confluent by 1 min treatment with 10 ml versene (Thermo Fisher Scientific) to remove the feeder cells, followed by 5 min treatment with trypsin/EDTA (Thermo Fisher Scientific). Dislodged keratinocytes were washed and re-plated at ~2500 keratinocytes/cm² with irradiated 3T3-J2 feeder fibroblasts.

## Tenofovir treatment of primary vaginal keratinocytes

Tenofovir (CAS 147127-20-6; T018500, Toronto Research Chemicals, Canada) was dissolved in phosphate-buffered saline, 7% dimethyl sulfoxide, and 5% 5N sodium hydroxide to result in a 767 mM stock solution, and further diluted in culture media for addition to keratinocyte cultures in concentrations ranging from 0.05 to 32 mM. Based on initial titration experiments in which we measured the intracellular concentration of tenofovir diphosphate, the active cellular metabolite of tenofovir, by liquid chromatography-tandem mass spectrometry (performed in the laboratory of Dr Craig Hendrix, Johns Hopkins University), concentrations in the culture media at the lower end (0.05–0.5 mM range) were estimated to be equivalent to the active concentrations likely achieved by topical tenofovir 1% gel (35 mM) in mucosal epithelial cells in vivo (*Hendrix et al., 2013*; *Louissaint et al., 2013*). In all experiments, control keratinocytes were cultured in parallel without tenofovir. Keratinocytes were harvested at pre-determined time points, split into several aliquots, and pelleted. Pellets were frozen and stored at −80°C either as dry pellets for protein assays or after suspension in RNAprotect Cell Reagent (Qiagen) for RNA assays.

## alamarBlue cell proliferation and cell viability assay

Primary vaginal keratinocytes were cultured in six-well plates (Corning) with or without various concentrations of tenofovir for up to 14 days. At day 1, day 4, day 7, or day 14, the alamarBlue reagent (BUF012A; AbD Serotec) was added. After 5 hr, absorbance was measured at 570 and 600 nm on a Varioskan Flash Multimode reader (Thermo Fisher Scientific). Percentage reduction of alamarBlue, corresponding to the level of cell proliferation, was calculated as specified in the manufacturer's technical datasheet. This was converted to percentage of maximal alamarBlue reduction by dividing each well by the well with the highest amount of alamarBlue reduction.

## Interleukin-10 ELISA assay

The IL-10 concentration in primary vaginal keratinocytes was measured after lysis of frozen cell pellets in Cell Lysis buffer (R&D Systems, Minneapolis, MN) using a commercial human IL-10 immunoassay (Quantikine HS, HS100C; R&D Systems) according to the manufacturer's specifications.

## RNA isolation, cRNA preparation, and whole genome BeadArray hybridization

RNA was isolated from the biopsies using the RNeasy Fibrous Tissue Mini Kit (Qiagen), and from the vaginal keratinocytes using the Direct-zol RNA MiniPrep kit (Zymo Research, Irvine, CA), according to the manufacturer's instructions, treated with 27 Kunitz units of DNAse (Qiagen) to remove genomic DNA contamination, and evaluated for integrity using the Agilent RNA 6000 Nano Kit (Agilent, Palo Alto, CA) on an Agilent 2100 Bioanalyzer. All samples had an RNA Integrity Number of 7 or greater. 500 ng of total RNA was amplified and labeled using the Illumina TotalPrep RNA Amplification kit (Thermo Fisher Scientific). cRNA from a total of 192 rectal biopsy samples (eight men per study arm, four study arms, three time points, and two biopsy sites [9 cm and 15 cm]), and from a total of 36 primary vaginal keratinocyte cultures (three tissue donors, three arms, and four time points), was hybridized to HumanHT12 v4 Expression BeadChips (Illumina, San Diego, CA) according to the manufacturer's protocols. Each chip contains 47,323 probes, corresponding to 30,557 genes.

## Unbiased mass spectrometry of rectal sponge eluates

Sponges were obtained on the same visits as the biopsies, prior to the biopsy procedure, and processed as previously described (*Anton et al., 2011*). Protein concentration of sponge eluates was determined by BCA assay (EMD Millipore). Equal amounts of total protein from each sample (100 µg) were then denatured in pH 8.0 urea exchange buffer (8 M Urea [GE HealthCare, United Kingdom], 50 mM HEPES [Sigma-Aldrich]) for 20 min at room temperature, and then placed into 10 kDa Nanosep filter cartridges. After centrifugation, samples were treated with 25 mM dithiothreitol (Sigma-Aldrich) for 20 min, then 50 mM iodoacetamide (Sigma-Aldrich) for 20 min, followed by a wash with 50 mM HEPES buffer. Trypsin (Promega, Fitchburg, WI) was added (2 µg/100 µg protein) and samples were incubated at 37°C overnight in the cartridge. Peptides were eluted off the filter with 50 mM HEPES, and the digestion was stopped with 1% formic acid. Peptides were dried via vacuum centrifugation and cleaned of salts and detergents by reversed-phase liquid chromatography (high pH RP, Agilent 1200 series micro-flow pump, Water XBridge column) using a step-function gradient. The fractions were then dried via vacuum centrifugation. Equal amounts of peptides were re-suspended in 2% acetonitrile (Thermo Fisher Scientific), 0.1% formic acid (EMD, Canada) and injected into a nano-flow LC system (Easy nLC, Thermo Fisher Scientific) connected in-line to a LTQ Orbitrap Velos mass spectrometer (Thermo Fisher Scientific). Mass spectrometry instrument settings were the same as described previously (*Burgener et al., 2013*).

## Quality control and processing of microarray and mass spectrometry data

### Microarray data

All chips used in the analysis passed standard quality control metrics assessed by GenomeStudio (Illumina) as well as visual inspection for anomalies and artifacts. GenomeStudio calculates a detection p value for each probe, which represents the confidence that a given transcript is expressed above background defined by negative control probes. Further processing and statistical analysis of data was done using the R/Bioconductor software suite (*Gentleman et al., 2004*). To aid in the comparison of gene expression data gathered across the series of hybridization chips used, data were normalized by variance stabilizing transformation and robust spline normalization as described in the Bioconductor lumi package (*Du et al., 2008*; *Lin et al., 2008*). The pre-processed data were filtered to include only probes with detection threshold p values of <0.05 in 100% of biopsies in at least one of the four study arms, or in 100% of vaginal epithelial cell cultures, and to remove probes that had a low standard deviation (≤0.5) across all arrays, using the Bioconductor genefilter package (*Bourgon et al., 2010*). Finally, probes without an Entrez ID were removed, leaving 1928 probes in the rectal biopsies, and 6699 probes in the vaginal epithelial cell cultures, for further statistical analysis.

### Mass spectrometry data

All spectra were processed using Mascot Distiller v2.3.2 (Matrix Science, Boston, MA), against the UniProtKB/SwissProt (2012-05) Human (v3.87) database using the decoy database option (2% false discovery rate), along with the following parameters: carbamidomethylation (cysteine residues) (K and N-terminus) as fixed modifications, oxidations (methionine residues) as a variable modification, fragment ion mass tolerance of 0.5 Da, parent ion tolerance of 10 ppm, and trypsin enzyme with up to 1 missed cleavage. Mascot search results were imported into Scaffold (v4.21) (Proteome Software, Portland, OR) and filtered using 80% confidence for peptides, 99% confidence for proteins, and at least two peptides per protein. Label-free protein expression levels based on MS peak intensities were calculated using Progenesis LC-MS software V4.0 (Nonlinear Dynamics, Durham, NC). Feature detection, normalization, and quantification were all performed using default settings in the software. Retention time alignment was performed using automatic settings and manually reviewed for accuracy on each sample. Only charge states between 2+ and 10+ were included. Protein abundances were further normalized by total ion current.

## Statistical analysis of microarray and mass spectrometry data

### Microarray data

A Bayesian probabilistic framework, Cyber-T (*Baldi and Long, 2001*; *Long et al., 2001*), was run from the CyberT/hdarray library in R to test whether differences in gene expression observed between before and after tenofovir treatment were significant. The effect of tenofovir 1% gel on the rectal mucosa in vivo was tested in paired comparisons between time point I or VII and time point 0.

Each treatment arm was considered separately, and researchers were blinded to treatment arm. The effect of tenofovir on vaginal epithelial cell cultures in vitro was tested in paired comparisons between treated and untreated cultures. Each tenofovir dosage (50 and 500 µM) and time point (1, 4, 7, and 14 days) was considered separately. The Benjamini & Hochberg method for estimating false discovery rates (FDR) was used to control for multiple comparisons (*Benjamini and Hochberg, 1995*). Criteria for significance and relevance were an estimated FDR $\leq 0.05$ and a $\log_2$ fold expression change of $\geq 0.5$ (induction) or $\leq -0.5$ (suppression), respectively. To confirm the results of the in vivo MTN-007 study, we reanalyzed the data using a double subtraction strategy, where the paired differences between time point 0 and time points I or VII within each treatment arm were first calculated for each probe and study participant. In a second step, the mean paired differences for each probe within each of the three treatment arms were compared to mean paired differences for each probe within the no-treatment arm. Significance testing for this alternative analysis of the in vivo MTN-007 data was performed via a linear fit model using the limma package (*Smyth, 2005*), with the same significance criteria used for Cyber-T. The resulting gene lists greatly overlapped with the Cyber-T results. For simplicity, the numbers of induced and suppressed genes reported in 'Results' are based on the Cyber-T analysis. Heat maps of differentially regulated genes were generated using MeV 4.8 within the TM4 Microarray Software Suite (*Saeed et al., 2006*), and hierarchically clustered according to selected gene ontologies found in the databases DAVID 6.7 (*Huang da et al., 2009*) and InnateDB (*Lynn et al., 2008*). All microarray data were deposited into the GEO database (accession numbers, GSE57025 and GSE57026).

### Mass spectrometry data

For each protein, $\log_2$ protein intensity values were averaged across all five subjects separately for time points 0 and VII, fold change values between the two time points were calculated, and statistical analysis for differences between the two time points was done by paired t tests.

## Pathway and network analysis of microarray and mass spectrometry data

### Microarray data

Entrez ID designations, assigned to the array probes by Illumina, were uploaded to the Innate DB database and a Gene Ontology (GO) over-representation analysis was performed for gene groups signifying a particular molecular function or biological process, or occurring in specific cellular compartments (*Lynn et al., 2008*). The following strategy was used to determine which gene groups were enriched in the data set: separately for suppressed and induced genes, ratios of the number of genes in a particular GO group to the total number of genes detected in our data set were compared to the ratios in the same GO group reported for the complete human genome using Fisher's exact test. Ingenuity Pathways Analysis (IPA) (Qiagen) was used to visualize direct and indirect relationships between individual gene products and map their cellular localizations.

### Mass spectrometry data

Proteomic data were uploaded into the IPA software package (Qiagen). Associations between protein groups in the data set and canonical pathways were measured similarly to the over-representation analysis described for the microarray data above. IPA software was also used to compare protein expression patterns against the IPA biological function database. IPA software calculates a Benjamini & Hochberg-adjusted p-value for the association of protein abundance changes with each biological function, and a weighted z-score, which estimates whether the protein abundance changes found associated with a specific biological function are likely to enhance (positive z values) or inhibit (negative) the function.

## Quantitative confirmation of microarray results by RT-ddPCR

A two-step reverse transcription (RT) droplet digital PCR (ddPCR) was used to confirm microarray transcriptome data for selected genes of interest (*Hindson et al., 2011*; *Pinheiro et al., 2012*). In a ddPCR assay, each sample is partitioned into ~20,000 droplets representing as many individual PCR reactions. The number of target DNA copies present per sample can be quantified based on Poisson distribution statistics, because each individual droplet is categorized as positive or negative for a given gene. For rectal biopsies, cDNA was generated using the High Capacity cDNA Reverse Transcription Kit (Thermo Fisher Scientific) and the ddPCR was carried out using the 2× ddPCR Supermix for Probes (BioRad, Hercules, CA). For epithelial cell lines, these steps were combined using

the 2× One-Step RT-ddPCR Kit for Probes (BioRad). ddPCR was performed on the QX100 droplet digital PCR system (BioRad). Reactions were set up with 20× 6-carboxyfluorescein (FAM)-labeled target gene-specific qPCR assay (Integrated DNA Technologies, Coralville, IA; or Thermo Fisher Scientific) and 20× VIC-labeled housekeeping hemoglobin B (HBB) gene-specific Taqman gene expression assay (Thermo Fisher Scientific). Each assembled ddPCR reaction mixture was loaded in duplicate into the sample wells of an eight-channel disposable droplet generator cartridge (BioRad) and droplet generation oil (BioRad) was added. After droplet generation, the samples were amplified to the endpoint in 96-well PCR plates on a conventional thermal cycler using the following conditions: denaturation/enzyme activation for 10 min at 95°C, 40 cycles of 30 s denaturation at 94°C, and 60 s annealing/amplification at 60°C, followed by a final 10 min incubation step at 98°C. After PCR, the droplets were read on the QX100 Droplet Reader (BioRad). Analysis of the ddPCR data was performed with QuantaSoft analysis software version 1.3.1.0 (BioRad).

## Immunohistochemistry of formalin-fixed rectal biopsies

Four-micron sections were cut on a Leica RM2255 Automated Rotary Microtome (Leica, Germany), mounted on positively charged EP-3000 slides (Creative Waste Solutions, Tualatin, OR), dried for 1 hr at 60°C, and stored until staining at 4°C. Slides were deparaffinized in xylene and rehydrated in graded dilutions of ethanol in water. Antigen retrieval was performed by heating the slides in Trilogy Pretreatment Solution (Sigma-Aldrich) for 20 min at boiling temperature in a conventional steamer (Black&Decker, Towson, MD). Slides were then cooled for 20 min, rinsed three times in Tris-buffered 0.15 M NaCl solution containing 0.05% Tween 20 (Wash Buffer; Dako, Santa Clara, CA), and stained at room temperature in an automated slide-processing system (Dako Autostainer Plus). Endogenous peroxidase activity was blocked using 3% $H_2O_2$ for 8 min, followed by 10 min in Serum-Free Protein Block (Dako). The slides were then stained for 1 hr with the primary antibodies. Staining was performed with the following primary antibodies: anti-CD3 (RM9107S, rabbit clone SP7; Thermo Fisher Scientific), anti-ubiquitin D (NBP2-13498, rabbit polyclonal anti-FAT10 antibody; Novus Biologicals, Littleton, CO), anti-CD7 (M7255, mouse clone CBC.37; Dako), and anti-interleukin-10 (sc-8438, mouse clone E−10; Santa Cruz Biotechnology, Dallas, TX). Negative control primary immunoglobulins were either whole rabbit IgG (011-000-003; Jackson ImmunoResearch Laboratories, West Grove, PA) or mouse IgG (I-2000; Vector Laboratories, Burlingame, CA).

For CD3 staining, the slides were incubated with anti-CD3 for 60 min at a dilution of 1:25, washed in Wash Buffer, incubated for 30 min with sheep-anti-rabbit Dylight 649 (611-643-122; Rockland Immunochemicals, Limerick, PA), washed, and incubated for 30 min with donkey-anti-sheep Dylight 649 (613-743-168; Rockland). Sections were counter-stained for 20 min with the nucleic acid-binding dye Sytox Orange (S11368; Thermo Fisher Scientific) at a dilution of 1:20,000, and coverslipped in ProLong Gold antifade reagent (P36930; Thermo Fisher Scientific). For ubiquitin D (UBD), CD7 and interleukin-10 (IL-10) staining, the slides were incubated with 0.8 µg/ml (UBD and CD7) or 4 µg/ml (IL-10) of the primary antibody for 60 min. After washing, the anti-UBD-stained slides were incubated for 30 min with Leica Power Vision HRP rabbit-specific antibody polymer (PV6119; Leica); and the anti-CD7 and anti-IL-10 stained slides were incubated for 30 min with LeicaPower Vision HRP mouse-specific antibody polymer (PV6114; Leica). After washing, staining was visualized with 3,3′-diaminobenzidine (Liquid DAB+ Substrate Chromogen System; Dako) for 7 min, and the sections were counter-stained for 2 min with hematoxylin (Biocare, Concord, CA). Controls for all antibodies were run with identical procedures but replacing the primary antibodies with either rabbit IgG or mouse IgG, as appropriate, at calculated matching concentrations.

## Acquisition and analysis of stained tissue sections

Sequences of slightly overlapping 20× (for CD3, UBD or CD7) or 40× (for IL-10) images covering each stained tissue section in its entirety were acquired on a bright-field Aperio Scansope AT (for UBD, CD7 and IL-10) or a fluorescent Aperio Scansope FL (for CD3) (Aperio ePathology Solutions, Leica). Overlapping images were stitched together using the Aperio Image Analysis Suite, and the entire tissue sections on each slide were analyzed using Definiens Tissue Studio (Definiens AG, Germany). Individual cells were identified based on the nuclear counter-stain. For CD3 and CD7, all positive cells were automatically counted and reported as number of positive cells per mm$^2$ of tissue section. For IL-10 and UBD, tight regions were manually drawn around all areas of the tissue sections containing columnar epithelial cells. For IL-10, all positive columnar epithelial cells were

automatically counted and reported as number of positive cells per mm². For UBD, which stained all epithelial cells, the mean staining intensity (MSI) of each columnar epithelial cell was measured in arbitrary units and overall UBD staining intensity for each section was reported as the average MSI of all epithelial cells.

## Electron microscopy

Formalin-fixed paraffin-embedded rectal biopsies were de-paraffinized and fixed overnight in half-strength Karnovsky's fixative. Staining, embedding, cutting, and viewing on a JEOL 1400 SX transmission electron microscope were performed as previously described (*Hladik et al., 1999*, *2007*). 20 images per sample were acquired at 5,000× magnification. Using ImageJ (*Collins, 2007*), the two-dimensional sizes in µm² of all individual mitochondria with a circularity index of ≥0.9 were calculated (for standardization purposes, only mitochondria cut near perfectly along their minor axis were evaluated). 10 images per sample were also acquired at 2,000× magnification, always including the epithelial cell brush border. Using ImageJ, a grid of 1.32 µm² squares of defined size was overlaid onto each image. All mitochondria in the images were counted, except those in the first row of squares falling on the brush border and in squares along the image rims (which only partially covered the tissue), and the mean numbers of mitochondria per µm² were calculated for each of the acquired 2,000× images (range of counted squares per image: 23–50).

## General statistics

All p values reported were adjusted for multiple testing as appropriate, except for the exploratory protein mass spectrometry data. Correlations of $\log_2$ fold gene expression changes between 9 cm and 15 cm biopsies in Figure 1C and Figure 1—figure supplement 1 were tested by Spearman's rank correlation coefficient. Gene and protein expression changes over baseline were compared between study arms by two-tailed Mann–Whitney test (Figure 1—figure supplement 2 and Figure 6). The combined expression data from the microarray and RT-ddPCR tests shown in Figure 2C were compared separately for log-fold induction or suppression over baseline using a one-tailed Wilcoxon signed-rank test with Bonferroni adjustment. Immunohistology measurements in Figure 2D were compared between baseline and after 7 days of treatment by two-tailed paired t tests. Due to high skewness of the CD7[+], CD3[+], and IL-10[+] cell counts, these were tested after $\log_{10}$ transformation. Ratios of induced to suppressed genes in Figure 3A were tested for a difference between cellular compartments using Chi-square statistics. DSP, IL-10, KIAA0101, PNPT1, and ATP6 copy numbers or protein concentrations in Figures 4B,C, 5A,B,F were tested for significant change across three time points by repeated measures ANOVA, and exact Sidak's (Figures 4B,C, 5F) or Tukey (Figure 5A,B) post-tests were run for comparisons between two time points. Due to high skewness of the KIAA0101 copy numbers, these were tested after $\log_{10}$ transformation. The effect of increasing tenofovir concentrations on the proliferation of primary vaginal epithelial cells in Figure 4E was assessed for significance by a linear model accounting for days and donors. Copy numbers between 9 cm and 15 cm biopsies in Figure 5A were compared by two-tailed paired t test. Copy numbers between baseline tenofovir and N-9 in Figure 5B were compared by a two-tailed unpaired t test. Mitochondria counts and sizes in Figure 5C,D were compared between baseline and Day VII, separately for the two subjects evaluated by electron microscopy, by two-tailed unpaired t tests with Bonferroni adjustment. Microarray chip probe expression values were tested for correlation between the three primary vaginal cell cultures by computing pairwise Pearson correlation coefficients (*Figure 3—source data 1*). Statistics packages used were Prism 6 (Graphpad Software, La Jolla, CA) and Bioconductor/Lumi in R. Statistical results are summarized in *Supplementary file 1*.

## Acknowledgements

We thank the study participants for their time and effort, and staff in the participating clinics for enrolling and following study participants. We thank Peter Wilkinson and Rafick-Pierre Sékaly from the Vaccine and Gene Therapy Institute of Florida and Sangsoon Woo from the Vaccine and Infectious Disease Division at the Fred Hutchinson Cancer Research Center for advice about microarray data analysis; Gustavo Doncel from CONRAD and the Eastern Virginia Medical School, Michael Boeckh and Stephen Voght from the Fred Hutchinson Cancer Research Center, and Anna Wald from the University of Washington, for critical reading and editing of the manuscript; Keith Jerome from the Vaccine and Infectious Disease Division at the Fred Hutchinson Cancer Research Center for assistance with ddPCR

assays; Julie Randolph-Habecker and Kim R Melton from the Experimental Histopathology and Bobbie Schneider from the Electron Microscopy Shared Resources Core Facility at the Fred Hutchinson Cancer Research Center for assistance with immunohistochemistry and electron microscopy; Cary A Moody from the Department of Microbiology and Immunology, University of North Carolina–Chapel Hill, for providing the 3T3-H2 fibroblast feeder cell line; Gretchen Lentz and Michael Fialkow from the Department of Obstetrics and Gynecology at the University of Washington for procuring vaginal tissues for the isolation of vaginal keratinocytes; Aornrutai Promsong and Surada Satthakarn, currently at Prince of Songkla University, Thailand, for help with establishing the four primary vaginal keratinocyte cultures; Allison A McBride from the Laboratory of Viral Diseases at the National Institute of Allergy and Infectious Diseases for advice regarding the culture of primary vaginal keratinocytes; and Craig W Hendrix and Mark A Marzinke from the Division of Clinical Pharmacology, Department of Medicine, The Johns Hopkins University School of Medicine, for measuring the intracellular concentration of tenofovir diphosphate in primary vaginal keratinocytes treated with tenofovir in vitro; Kenzie Birse from the Department of Medical Microbiology, University of Manitoba, for support in proteomic data analysis; Max Abou, Garrett Westmacott, and Stuart McCorrister of the Proteomic Core of the National Microbiology Laboratory at the Public Health Agency of Canada for proteomic sample preparation and technical assistance in mass spectrometry.

## Additional information

### Funding

| Funder | Grant reference number | Author |
| --- | --- | --- |
| National Institutes of Health (NIH) | U01AI068633 | Ian McGowan |
| National Institutes of Health (NIH) | U19AI082637 | Ian McGowan |
| National Institutes of Health (NIH) | R01HD51455 | Florian Hladik |

The funder had no role in study design, data collection and interpretation, or the decision to submit the work for publication.

### Author contributions

FH, IMG, Conception and design, Analysis and interpretation of data, Drafting or revising the article; AB, LV, JS, TBB, Acquisition of data, Analysis and interpretation of data; LB, Acquisition of data, Analysis and interpretation of data, Drafting or revising the article; RG, SF, JYD, MJC, Analysed gene expression data, Analysis and interpretation of data; SMH, Analysis and interpretation of data, Drafting or revising the article; CH, Recruitment of study participants, Acquisition of data; PA, SJ, Operational support for the clinical study, Acquisition of data; JP, Advised on all aspects of the study, Conception and design; DRF, Provided study drug, Contributed unpublished essential data or reagents; RDC, Recruitment of study participants, Drafting or revising the article; KHM, Recruitment of study participants, Acquisition of data, Drafting or revising the article; MJME, Conception and design, Drafting or revising the article

### Author ORCIDs

Florian Hladik, http://orcid.org/0000-0002-0375-2764

### Ethics

Clinical trial Registry: NCT. Registration ID: NCT01232803.
Human subjects: MTN-007 (ClinicalTrials.gov registration NCT01232803) was a phase 1, double blind, placebo-controlled trial in which participants were randomized to receive rectal reduced glycerin tenofovir 1%, nonoxynol-9 (N-9) 2% or hydroxyethylcellulose (HEC) gels, or no-treatment (1:1:1:1), at three clinical research sites (Pittsburgh, PA; Birmingham, AL; and Boston, MA). The study protocol was approved by IRBs at all three sites. All participants gave written informed consent. The study details, and general safety and acceptability data, have been published elsewhere (McGowan I et al. [2013] A phase 1 randomized, double blind, placebo controlled rectal safety and acceptability study of tenofovir 1% gel [MTN-007]. PLOS ONE 8: e60147). The CONSORT checklist and CONSORT flow chart are provided.

## Additional files

### Supplementary files

• Supplementary file 1. Summary of effect sizes and statistical significance testing.

• Supplementary file 2. Lists of genes significantly up- or down-regulated by tenofovir 1% or nonoxynol-9 2% gels in 9 cm rectal biopsies over 7 days of treatment.

• Reporting standard 1. CONSORT checklist.

• Reporting standard 2. CONSORT flow chart.

### Major datasets

The following datasets were generated:

| Author(s) | Year | Dataset title | Dataset ID and/or URL | Database, license, and accessibility information |
| --- | --- | --- | --- | --- |
| Fleming L, Hladik F | 2014 | MTN_007 Data | GSE57025; http://www.ncbi.nlm.nih.gov/geo/query/acc.cgi?acc=GSE57025 | Publically available at the NCBI Gene Expression Omnibus (http://www.ncbi.nlm.nih.gov/geo/). |
| Fleming L, Hladik F | 2014 | Ex vivo TFV data | GSE57026; http://www.ncbi.nlm.nih.gov/geo/query/acc.cgi?acc=GSE57026 | Publically available at the NCBI Gene Expression Omnibus (http://www.ncbi.nlm.nih.gov/geo/). |

Standard used to collect data:

The clinical trial recruiting subjects for the current paper was published in McGowan I et al. (2013) A phase 1 randomized, double blind, placebo controlled rectal safety and acceptability study of tenofovir 1% gel (MTN-007). PLOS ONE 8: e60147.

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
