## [Decision Letter]

Thank you for sending your work entitled “Mucosal effects of tenofovir 1% gel” for consideration at *eLife*. Your article has been favorably evaluated by Prabhat Jha (Senior editor), Sema Sgaier (Reviewing editor), and 3 reviewers, one of whom, Stephen Becker, has agreed to share his identity.

The Reviewing editor and the reviewers discussed their comments before we reached this decision, and the Reviewing editor has assembled the following comments to help you prepare a revised submission.

The Reviewing editor and the reviewers agree that this work is very important, timely, comprehensive, and should be published. However, before making a final decision, we would like the minor comments below to be addressed.

1) It appears that the data reported here are from the use of the reduced glycerin rectal formulation of tenofovir gel. Since there is testing on vaginal mucosa, there may have also been use of the standard formulation. Please ensure that it is clear to the reader which formulation has been used and include in the Discussion whether the authors feel that the findings are pertinent to the formulation examined in efficacy studies.

2) Methods section:

a) Paragraph 12 is a bit confusing with regard to understanding which specimen type (vaginal keratinocytes or rectal mucosa) is being tested. It appears to be rectal tissue because of the reference to time point 0, I and VII, however the description is in a paragraph that also discusses testing of the vaginal tissue.

b) Rectal study:

Please clarify when the rectal sponge samples were collected (at the same time as the biopsies?).

Were the samples blinded to the researchers?

The 1-day and 7-day treatment samples not only differed in treatment duration, but also in timing of sample collection post treatment. The 1-day samples were collected 30-60 min after tenofovir application and would have shown only acute effects, whereas the 7-day samples were collected 24 hours after the last dose and would have shown chronic effects. To more accurately compare differences in gene expression between one-dose vs 7-daily doses, the same post treatment time point(s) should be used for sampling. The significance of the difference in sample timing at the two treatment time points should be clarified in the text.

c) Vaginal cell experiments:

Please explain the use of cholera toxin and hydrocortisone in the media, and the effects they may have on inflammatory endpoints?

If possible, provide more details concerning the vaginal tissue donors (age, hormonal use etc.), and quality control measures used for cell cultures (number of passages, lack of contamination with fibroblast or other cell types etc).

Please explain the significance of the Alamar blue data shown in the supplement.

3) Discussion section:

In the fourth paragraph it states that while KIAA0101 and UBD appear to be affected by tenofovir gel, there is no actual clinical evidence for carcinogenicity. Does this refer to lack of evidence from the data presented here? If so, could it be argued that the exposure/follow up here is sufficient to make such a statement? If based on prior data, such as animal carcinogenesis studies performed as part of preclinical development of tenofovir, this is should be clarified and the studies should be cited.

---

## [Author Response]

*1) It appears that the data reported here are from the use of the reduced glycerin rectal formulation of tenofovir gel. Since there is testing on vaginal mucosa, there may have also been use of the standard formulation. Please ensure that it is clear to the reader which formulation has been used and include in the Discussion whether the authors feel that the findings are pertinent to the formulation examined in efficacy studies*.

The vaginal mucosa was not tested; rather, we isolated primary vaginal epithelial cells from four healthy women and exposed them to tenofovir in vitro. These tenofovir exposures were not done in gel form but as tenofovir dilutions in media. We have revised the Abstract to make this clear.

*2) Methods section*:

*a) Paragraph 12 is a bit confusing with regard to understanding which specimen type (vaginal keratinocytes or rectal mucosa) is being tested. It appears to be rectal tissue because of the reference to time point 0, I and VII, however the description is in a paragraph that also discusses testing of the vaginal tissue*.

We think the confusion arose from the different nature of in vivo (MTN-007) and in vitro (epithelial cells) data. In vitro, we can treat the same cells with tenofovir or leave them untreated. Therefore, direct pairwise comparisons between treated and untreated cells are possible. In vivo, each subject only received one treatment, and statistical analysis of the in vivo data has to be done in a longitudinal fashion as pairwise longitudinal comparisons between before and after treatment in each subject. In contrast to the more straight forward analysis of the in vitro data, the in vivo data can therefore be analyzed in two different ways, as described in the Methods.

In the Methods section for statistical microarray analysis, we have now clarified that only the in vivo MTN-007 data were analyzed in two ways, whereas the in vitro data were analyzed in one way.

*b) Rectal study*:

*Please clarify when the rectal sponge samples were collected (at the same time as the biopsies?)*.

The rectal sponges were at the same visits, immediately before the biopsies were taken. We added this information to the Methods section “Unbiased Mass Spectrometry of Rectal Sponge Eluates”.

*Were the samples blinded to the researchers*?

Yes. Unblinding of the microarray results was in fact the most surprising moment of the study. We knew (since we did have time point information) that one arm showed a particularly large number of gene expression changes. Based on our initial hypotheses we were convinced that this had to be the nonoxynol-9 (N-9) arm. When it turned out to be the tenofovir arm, we at first thought that a mistake had occurred during unblinding. However, further cross-checking confirmed that 1% tenofovir led to many more changes than 2% N-9. This was borne out by the subsequent in-depth pathway analysis, which revealed mitochondrial changes that are typical for NRTIs but not for N-9, and subsequently the good concordance between the in vivo and the in vitro tenofovir data.

*The 1-day and 7-day treatment samples not only differed in treatment duration, but also in timing of sample collection post treatment. The 1-day samples were collected 30-60 min after tenofovir application and would have shown only acute effects, whereas the 7-day samples were collected 24 hours after the last dose and would have shown chronic effects. To more accurately compare differences in gene expression between one-dose vs 7-daily doses, the same post treatment time point(s) should be used for sampling. The significance of the difference in sample timing at the two treatment time points should be clarified in the text*.

We have now emphasized this difference better in the text of the Methods section “Mucosal Biopsy Procedures”.

*c) Vaginal cell experiments*:

*Please explain the use of cholera toxin and hydrocortisone in the media, and the effects they may have on inflammatory endpoints*?

This media formulation is standard for culturing primary epithelial cells since publication of a classic paper in Cell by Rheinwald and Green (Cell 6:331-344; 1975), and many follow-up papers by Jim Rheinwald (Harvard) and others. However, none explains mechanistically why these two particular ingredients are needed. It is true that cholera toxin could be considered as potentially pro-inflammatory and hydrocortisone as potentially anti-inflammatory. However, they were present in all cultures, and therefore their effect should not contribute to differences between tenofovir-treated and -untreated cultures. The two factors could perhaps lead to some changes longitudinally over time, but this is not tested in the in vitro experiments. In vitro, we compare tenofovir-treated and -untreated cells independently at each time point, but we do not compare between different time points.

*If possible, provide more details concerning the vaginal tissue donors (age, hormonal use etc), and quality control measures used for cell cultures (number of passages, lack of contamination with fibroblast or other cell types etc.)*.

We harvest discarded vaginal tissues under a waiver of consent, which prohibits us from obtaining any clinical or identifying information from the tissue donors. All we know is that the vaginal surgeries are performed for benign conditions. The strong concordance of the raw gene expression data between the three vaginal cell lines tested by microarrays (Figure 3–source data 1) indicates that unknown conditions in the tissue donors, or unknown differences of cell cultures in vitro, did not lead to identifiable outlier results in the in vitro experiments.

*Please explain the significance of the Alamar blue data shown in the supplement*.

The AlamarBlue data are part of Figure 4 (Figure 4), not the supplement, and indicate that tenofovir enhanced vaginal cell proliferation in vitro.

*3) Discussion section*:

*In the fourth paragraph it states that while KIAA0101 and UBD appear to be affected by tenofovir gel, there is no actual clinical evidence for carcinogenicity. Does this refer to lack of evidence from the data presented here? If so, could it be argued that the exposure/follow up here is sufficient to make such a statement? If based on prior data, such as animal carcinogenesis studies performed as part of preclinical development of tenofovir, this is should be clarified and the studies should be cited*.

Lack of carcinogenicity cannot be inferred from the short-term data in our study. No peer-reviewed studies exist that demonstrate carcinogenicity for tenofovir, nor do studies exist that reliably rule out carcinogenicity.